# CryoEM of Viral Ribonucleoproteins and Nucleocapsids of Single-Stranded RNA Viruses

**DOI:** 10.3390/v15030653

**Published:** 2023-02-28

**Authors:** Andrea Modrego, Diego Carlero, Rocío Arranz, Jaime Martín-Benito

**Affiliations:** 1Departamento de Estructura de Macromoléculas, Centro Nacional de Biotecnología Consejo Superior de Investigaciones Científicas (CNB-CSIC), 28049 Madrid, Spain; 2Departamento de Física de la Materia Condensada, Universidad Autónoma de Madrid (UAM), 28049 Madrid, Spain

**Keywords:** cryogenic electron microscopy (cryoEM), single-stranded RNA virus (ssRNAv), ribonucleoprotein (RNP), nucleoprotein (NP), virus assembly, genome packaging

## Abstract

Single-stranded RNA viruses (ssRNAv) are characterized by their biological diversity and great adaptability to different hosts; traits which make them a major threat to human health due to their potential to cause zoonotic outbreaks. A detailed understanding of the mechanisms involved in viral proliferation is essential to address the challenges posed by these pathogens. Key to these processes are ribonucleoproteins (RNPs), the genome-containing RNA-protein complexes whose function is to carry out viral transcription and replication. Structural determination of RNPs can provide crucial information on the molecular mechanisms of these processes, paving the way for the development of new, more effective strategies to control and prevent the spread of ssRNAv diseases. In this scenario, cryogenic electron microscopy (cryoEM), relying on the technical and methodological revolution it has undergone in recent years, can provide invaluable help in elucidating how these macromolecular complexes are organized, packaged within the virion, or the functional implications of these structures. In this review, we summarize some of the most prominent achievements by cryoEM in the study of RNP and nucleocapsid structures in lipid-enveloped ssRNAv.

## 1. Introduction

Over the last several years, cryogenic electron microscopy (cryoEM) has become a key method for the structural determination of a wide range of proteins and macromolecular assemblies with different sizes, and with an ever-increasing resolution [1]. This methodology in its two main facets, single particle analysis (SPA) and cryogenic electron tomography (cryoET), has been broadly applied in structural virology, providing a large number of valuable structures. According to the Electron Microscopy Database (www.emdataresource.org, accessed on 20 February 2023) around 6000 structures of viral proteins, complexes, and complete viruses have been deposited to date (Oct-2022), making cryoEM the technique that probably has contributed most to the resolution of viral structures in the last few years. With resolutions ranging from less than 2 Å to a few nm, these studies have not only allowed the building of more than 3000 atomic structures, shedding light on structural aspects of the virus assembly but have also revealed mechanisms of the dynamic processes carried out by viruses during infection. For example, cryoEM of isolated viral proteins, such as spikes or polymerases, has been used to determine the mechanism of inhibition of infection by antibodies against SARS-CoV-2 [2] or complex processes such as mRNA synthesis by influenza A polymerase [3,4].

Importantly, one of the main advantages of cryoEM is its ability to study large macromolecular assemblies, even in the case of high conformational variability, as well as to determine the structure of the reaction intermediates of biological processes carried out by these complexes. In this scenario, perhaps one of the fields in which this technique can be most useful is in the study of the main component that forms the viral nucleocapsid in some single-stranded RNA viruses: ribonucleoproteins.

## 2. The Single-Stranded RNA Viruses and Ribonucleoproteins

Single-stranded RNA viruses (ssRNAv) are a major cause of zoonotic outbreaks and therefore pose a great challenge in the control of infectious diseases affecting humankind. The fight against these pathogens must confront their great biological diversity and capacity to adapt to different hosts [5]. On this basis, the structural characterization of these viruses allows us to understand the mechanisms involved in viral infection and proliferation and, consequently, the possibility of developing new methods to combat such pathogens.

The ssRNAv can be divided into two groups: those whose genome is of positive polarity (+ssRNAv), which can be directly transcribed by host ribosomes to produce viral proteins; those whose genome is of negative polarity (-ssRNAv), in which the genome must first be copied into its complementary strand before it can be translated by the ribosomes [6]. In practically all -ssRNAv, and a few +ssRNAv, the genomic material forms a complex with multiple copies of a protein, usually basic in nature, known as nucleoprotein (N or NP), to which RNA binds in a nonspecific (not sequence-dependent) manner. These complexes are known as ribonucleoproteins (RNPs). In the case of -ssRNAv, in addition to the NP and genomic RNA, the RNP incorporates an RNA-dependent RNA-polymerase (RdRp) that is responsible for copying the genome to produce the positive polarity messenger RNA (mRNA) [7].

RNPs are the most complex component of ssRNAv and perform tasks crucial to their life cycles such as transcription, replication, and packaging of the genome to form progeny viruses; in other words, they are essential in all processes involving viral RNA. The ssRNAv may contain one or several RNPs, depending on whether their genome is segmented or not, which are densely packed inside the virion forming what is known as the nucleocapsid [8]. After infection, RNPs can also be found in the cytoplasm or nucleus of the host, since in many cases the incoming RNPs retain their structure to exert their functions. They can also be observed in the cell at late times post-infection due to the presence of de novo assembled RNPs, which encapsulate newly synthesized genomes that will form progeny viruses. It is also important to note that in some cases RNPs can behave as automatic RNA synthesis machines since, even in isolation, they can produce RNA with the mere presence of a primer and nucleotides [9,10].

This review aims to compile some of the most relevant cryoEM studies on the RNPs and nucleocapsids of ssRNAv, either of positive or negative polarity, including their organization within the virion and the functional implications of these structures.

## 3. Structure of the Nucleocapsid of -ssRNAv

As mentioned, viruses with genetic material composed of -ssRNA must incorporate within its nucleocapsid at least one copy of RdRp, whose function, as the first critical event after infection, is to copy the genome into a positive-polarity mRNA that can be translated by the ribosomes. The presence of this polymerase as a structural element in the virion is a distinctive feature of these viruses compared with +ssRNAv. Another distinctive feature is that during infection, the RNPs maintain their structure almost unaltered inside the host cell since it is a complete unit that is responsible for producing viral mRNA. The -ssRNAv can be divided into two groups solely according to structural criteria: viruses whose entire genome is contained on one RNA fragment, and thus form a single RNP; and those whose genome is fragmented in several segments, and therefore form multiple RNPs. Finally, it should be noted that although RNPs are organized into helical structures in every well-studied -ssRNAv, there are important structural differences between those with segmented and non-segmented genomes.

### 3.1. Nucleocapsids of Non-Segmented, -ssRNAv

The order containing non-segmented -ssRNA viruses is the *Mononegavirales*, which is made up of several families, among which the most important are the *Filoviridae*, *Rhabdoviridae*, *Paramyxoviridae*, and *Bornaviridae*. The genomes of these viruses can vary in size from around 9 to 19 kilobases (kb), with short terminal noncoding regions [11]. They encode a linear sequence of five to ten genes, slightly overlapping in some cases, with four core genes that are shared by all members: 1. nucleoprotein (N or NP); 2. phosphoprotein (P or VP35), a co-factor essential for transcription and replication in non-segmented viruses that has no direct counterpart in segmented ones [12]; 3. matrix protein (M); 4. large protein (L), which has RdRp activity [7,13].

From a structural point of view, these viruses stand out for having the most highly ordered helical nucleocapsids of all ssRNAv, and they also show a close interaction with the matrix protein that coats, also in a highly organized manner, the inner face of the virion membrane.

#### 3.1.1. Filoviridae: Ebola Virus (EBOV) and Marburg Virus (MARV)

EBOV and MARV viruses are enveloped and filamentous viruses that cause hemorrhagic fever and belong to the *Filoviridae* family (order: *Mononegavirales*) [14]. Their RNA genomes of around 18 and 19 kb, respectively, carry seven main genes [15] and are encapsidated by NPs and other viral proteins (VP30 and VP35) forming a highly regular helical nucleocapsid that serves as a scaffold for viral RNA synthesis [16].

Several cryoEM 3D reconstructions of the EBOV nucleocapsid have been obtained at different resolutions, providing detailed data on the structure of the components and the organization of the nucleocapsid complex. This data has informed hypotheses regarding protein functions during nucleocapsid formation and the viral life cycle. The earliest such study was performed using cryoET on native virions fixed in a 4% paraformaldehyde solution, and it produced the first three-dimensional (3D) reconstruction at a resolution of about 4 nm [17]. Later, Wan et al. [18] also employed cryoET and subtomogram averaging to determine the EBOV nucleocapsid structure at resolutions better than 1 nm (Figure 1A), and was performed either on nucleocapsid component proteins overexpressed in eukaryotic cells or by direct analysis of native viruses. Overexpression of a mutant NP containing only the first 450 amino acids (out of a total of 739 that make up the wildtype protein) produced a 50-nm-diameter helical structure bound to the host RNA, that was solved at 6.6 Å resolution. It was determined that the RNA was bound to a cleft formed by the lobes from the N- and C-terminal NP domains, with a stoichiometry of approximately six nucleotides per monomer. Joint overexpression of NP, VP35 (EBOV P protein), VP24 (a protein unique to filoviruses), and VP40 (EBOV M protein) also produced a large, 220-nm helical structure, whose structure was solved at 6.6 Å resolution. The organization of this complex showed that the RNA-bound NP helix was surrounded by helical layers formed by the other overexpressed proteins. Additionally in this work, cryoET of fixed native EBOV and MARV revealed a nucleocapsid with a structure very similar to that obtained by the joint overexpression of NP, VP35, VP24, and VP40 (Figure 1A–C,E–G). A subsequent study using SPA of this same helix formed by the overexpression of an NP 1-450 truncated mutant bound to host RNA produced a structure at 3.6 Å resolution, giving much more detailed information on NP-NP and NP-RNA interactions [19]. Other authors have since shown that overexpression of different EBOV NP fragments also leads to the formation of different helical structures, showing the enormous plasticity of this protein [20] (Figure 1D).

Similarly, the first studies on MARV were performed using cryoET, showing the helical organization of the native nucleocapsid at resolutions between 3 and 4 nm, both in isolated viruses and in viruses budding from infected cells [21]. Recently, the helical structure formed by overexpressed NP interacting with host RNA has allowed structural determination of its first 395 amino acids at 3.1 Å resolution [22] (Figure 1H). This helix, despite being different from that found in the nucleocapsid of virions, served as a basis for the study of how the RNA interacts with the NP, inspiring hypotheses regarding the mechanism of nucleocapsid formation and confirming the great structural similarity between EBOV and MARV.

**Figure 1 viruses-15-00653-f001:**
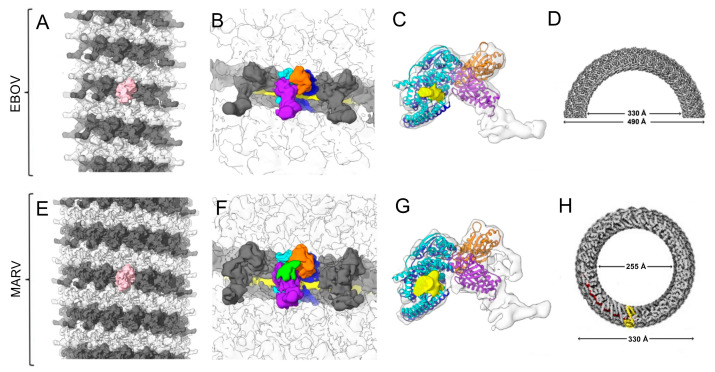
CryoEM structure of ebolavirus (EBOV) and Marburg virus (MARV) nucleocapsids from intact virions. (**A**) Overview of the EBOV nucleocapsid (NC). For ease of visualization, adjacent rungs are shown in dark and light gray, and a single subunit is highlighted in pink. The structure has 11.9 subunits per turn. (**B**) Segmentation of an EBOV NC subunit; two adjacent nucleoproteins (NPs) are shown in blue and cyan, two VP24 are shown in orange and purple, and the putative density corresponding to the RNA is shown in yellow. (**C**) Docking of the atomic structure of NP (blue and cyan) and VP24 (orange and purple) in the cryoEM map [18]. (**D**) Top view of another helical structure formed by overexpression of EBOV NP [20]. The structure has 42.4 NP subunits per turn. (**E**) Overview of the MARV NC; colors follow the same criteria as in A. The structure has 14.8 subunits per turn. (**F**) Segmentation of a subunit of the MARV NC; colors follow the same criteria as in B, and the green region is an extra disordered density present in the MARV NC. (**G**) Docking of the NP and VP24 atomic structures; colors follow the same criteria as in C [18]. (**H**) Top view of the helical structure of the NP-RNA complex of MARV; an NP monomer is highlighted in yellow and the RNA chain is highlighted in red [22]. The structure has 30.50 NP subunits per turn.

Additional reviews with more specific information related to the nucleocapsid and RNP structure of viruses within this family can be found in [23,24].

#### 3.1.2. Rhabdoviridae: Rabies Virus (RABV) and Vesicular Stomatitis Virus (VSV)

Within the order of the *Mononegavirales* is also the family *Rhabdoviridae*, whose members share a common elongated, rod-like, or bullet-like shape, which is distinctive from other viruses of this order [11]. Although the cell biology within the *Rhabdoviridae* family is quite divergent, they share a highly similar genetic and structural organization [25]. Their genomes are between 11 and 15 kb and encode six to eight proteins, four of which are usually part of the virion. The nucleocapsid consists of the RNP, comprising the genomic -ssRNA tightly bound to the N protein together with the RdRp polymerase (L) and phosphoprotein (P) [26].

VSV and RABV are two of the best structurally characterized members of this family, both with the typical bullet shape and genomes of 11 and 12 kb, respectively, which encode five proteins that are all present in virions. The first cryoEM 3D reconstruction of VSV was performed on native virions using SPA applied to the helical region of the virus’s main body [27]. The structure was determined at a resolution of 10.6 Å (Figure 2A), revealing that the nucleocapsid is formed by an internal helix containing the N protein bound to the genomic RNA, i.e., the RNP, tightly bound to another external helix formed by the matrix protein (M). The combination of these cryoEM data with the existing atomic structures of different fragments of the N and M proteins, as well as data from mutagenesis studies, allowed the proposal of an assembly mechanism for the virus. Recently, Jenni et al. [28] determined the structure at a higher resolution (3.5 Å), also from infectious virions. This work produced more detailed descriptions of the nucleocapsid structure, in which it was seen that the RNP could form helices with slightly different parameters in different virions and that the RNP was surrounded by two concentric layers of M protein, also organized into a helical structure (Figure 2B). Likewise, the RNP domain making up the dome that conforms the tip of the virion was resolved at about 9 Å, showing that it is formed by about 8 turns of a helix, with a diameter that increases until it reaches the regular diameter of the main trunk (Figure 2C). The final analysis allowed the authors to suggest a mechanism for membrane-coupled self-assembly of VSV that starts at the apex of the virion tip, with the 3′ end of the RNP forming an initial lock washer-shaped ring composed of about 10 N proteins, which grows into a helix of increasing diameter that will build the virion. This mechanism had previously been hinted at by other authors [27,29]. Another study performed by cryoET and sub volume averaging determined that there are approximately 50 copies of L protein (RdRp) included in each virion without following the helical symmetry, and they bind to the innermost part of the RNP by interacting with two consecutive N proteins [30]. Finally, a recent comprehensive study of VSV virions by cryoEM has demonstrated the interaction of the G protein (spike) with the M protein, deepening our understanding of virion assembly [31].

Despite its morphological and structural similarity, RABV has been studied less than VSV. Early cryoET studies exhibited the first data of native virions with a VSV-like arrangement. Notably, only one-third of the virions studied in this work had the canonical bullet shape, the rest of which were rounded or slightly elongated [32]. More recently, Riedel et al. [33] performed studies by cryoET and sub volume averaging on mutant RABV lacking the spicule (G) protein, revealing the nucleocapsid structure at a resolution of 15 Å with an overall organization similar to VSV, including an internal helical RNP containing the N protein and genomic RNA, surrounded by a layer formed by M protein that also has a helical structure (Figure 2D). This work also reported a partially pleomorphic behavior for this virus, in which virions of different shapes appeared.

**Figure 2 viruses-15-00653-f002:**
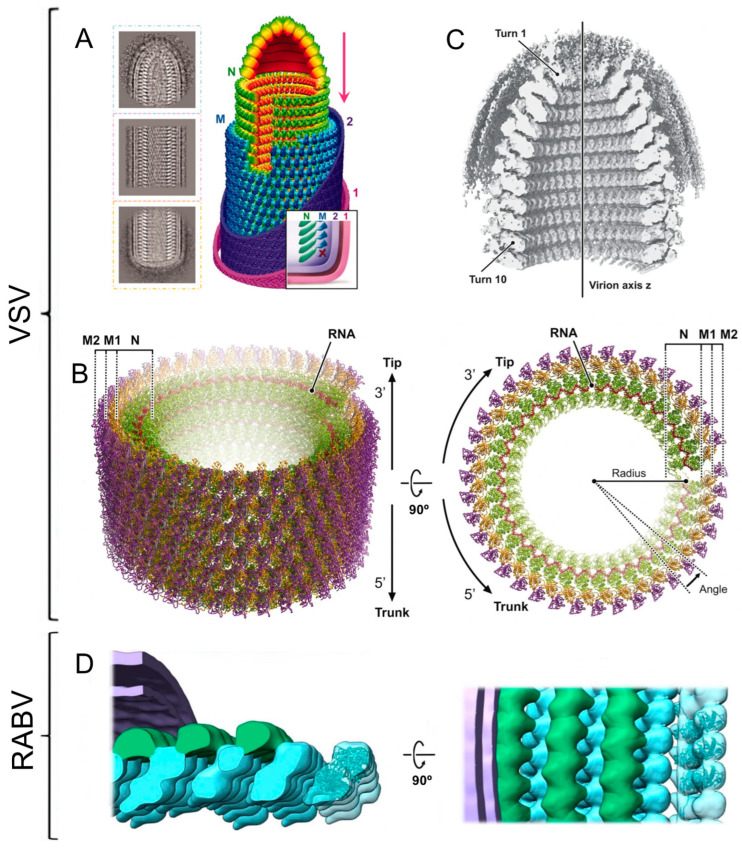
CryoEM structure of the vesicular stomatitis virus (VSV) and rabies virus (RABV) nucleocapsids. (**A**) 2D averages of the conical tip, trunk, and base of VSV and a full 3D model of the virion. The nucleoprotein (N) is depicted in red-yellow-green according to radial distance to the center, the matrix protein (M) in light-dark blue (according also to the radial distance), and the inner (2) and outer (1) membrane leaflets in purple and pink, respectively [27]. (**B**) Helical nucleocapsid structure with 38.5 N monomers per turn in lateral and top views. The N protein is colored green, the RNA in red, and the two layers of matrix protein (M1 and M2) are colored orange and purple, respectively. (**C**) Reconstruction of the VSV nucleocapsid tip without the imposition of any symmetry. To show the interior, the tip is cut in half [28]. (**D**) Two orthogonal views of the RABV nucleocapsid reconstruction. The densities corresponding to N, M, and membrane proteins are colored in cyan, green, and purple respectively. The atomic structure of three N proteins (PDB: 2GTT, cyan) are docked in the electron density map [33].

Other structures of rhabdoviral N-RNA complexes resolved by X-ray crystallography have been published [34]. Additional reviews based on the structure of the nucleocapsid and N protein of rhabdoviruses are in [35,36].

#### 3.1.3. Paramyxoviridae: Mumps Virus (MuV) and Measles Morbillivirus (MeV)

The *Paramyxoviridae* family is also part of the order *Mononegavirales*. These enveloped, pleomorphic viruses, although generally spherical in shape [11], have a genome of about 15 kb that encodes seven to ten proteins [26,37]. The helical RNPs of these viruses have a diameter of only 15 to 19 nm, with generally greater flexibility than those described so far in this review. It is important to note that flexibility, despite acting as a limitation for the achievement of high resolution by cryoEM techniques, is an important factor in the functionality of RNPs, as will be shown in other cases such as the segmented -ssRNAv.

The first studies on MeV RNPs were carried out using cryoET of native virions at 44 Å resolution [38]. Analysis of virion tomograms showed helical structures of two different diameters within virions (20 and 30 nm), with the larger-diameter structures being much more abundant (Figure 3A). This, together with other data, led the authors to hypothesize that the matrix (M) protein formed helical structures coating the RNP helix, rather than coating the inner face of the viral envelope [38] (Figure 3B). Subsequently, it was observed that overexpression of measles N protein produces RNP-like particles due to nonspecific binding to RNA present in the host cell [39,40]. The interaction of full-length N protein with RNA produces recombinant RNPs with a very flexible helical structure that does not allow high-resolution studies. However, a controlled N protein proteolysis by trypsin (resulting in the removal of the final 133 C-terminal amino acids) generates a much more rigid helix that is easier to study. In this way, Gustche et al. determined the structure of the recombinant MeV RNP-like particles at 4.3 Å resolution formed by what is called the Ncore protein (comprising amino acids 1-391). The N protein structure showed how RNA interacts through a groove between the N- and C-terminal domains, thus remaining on the outer side of the helix. The position of the RNA follows what is known as the “rule of six”, in which six nucleotides interact with each N protein, but with three bases pointing into the RNA binding groove and three bases pointing away from the cleft. Subsequent studies using this same MeV N protein fragment looking at interaction with specific 6-nucleotide RNA sequences, a poly-A, and the 5’-end sequence of the genomic RNA, produced a helical reconstruction at 3.3 Å resolution [41] (Figure 3C). This work also showed that the 3’ end of the genomic RNA is largely exposed in the RNP to enable interaction with the RdRp, thus driving a hypothesis about the mechanism of transcription initiation in vivo [41].

MuV is another paramyxovirus with a similar organization to MeV. In this case, studies have been performed on RNPs directly isolated from infectious virions at a resolution of 18 Å, revealing a helical structure of about 220 Å in diameter in which N protein is positioned to leave the RNA interaction cleft exposed on the external face of the helix [42]. The structure resulting from the interaction of the RNP with the C-terminal domain of the P protein was also determined to have helical parameters very similar to free RNP, but with an extra density (attributable to P protein) protruding on the external side of the helix [42]. Overexpression of the MuV N protein also produced different types of molecular aggregates upon its interaction with host cell RNA (Figure 3D). Shan et al. [43] obtained rings of 13 and 14 monomers as well as recombinant RNP-like particles in two helical conformations and resolving the structures at 3 to 5 Å (Figure 3E) allowed them to demonstrate the great structural plasticity of these proteins and hypothesize a mechanism for protein N polymerization.

**Figure 3 viruses-15-00653-f003:**
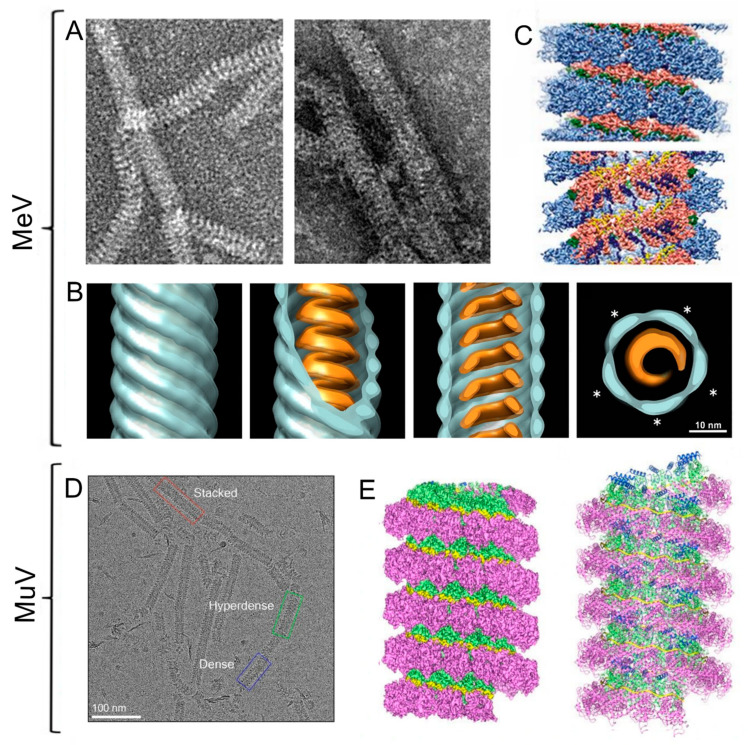
CryoEM structure of measles morbillivirus (MeV) and mumps virus (MuV). (**A**) Representative electron micrographs of MeV-infected (left) or nucleoprotein (N)- and matrix (M)-cotransfected (right) cell lysates prepared by immunosorbent electron microscopy. (**B**) Isosurface representations of the MeV nucleocapsid structure viewed laterally (first three images), and a slice taken across the axis (final image). The outer and inner parts (blue and orange, putatively M and N) show the helical twist, and the stars represent the five-star helical arrangement [38]. (**C**) Nucleocapsid-like helical particle structure obtained from the interaction of overexpressed MeV N protein in the presence of RNA. A side view (above) and a cutaway view (below) are shown [41]. (**D**) Representative cryoEM micrograph of helical structures obtained from overexpressed MuV N protein interacting with RNA; three different arrangements termed dense, hyperdense, and ring-stacked filaments are highlighted in blue, green, and red boxes, respectively. (**E**) 3D reconstruction of the dense helical filament and the respective atomic model [43].

Other RNP-like particle structures have been solved in recent years by cryoEM for other paramyxoviruses (e.g., a cetacean morbillivirus or Sendai virus) [44,45]. Additional reviews specific to the nucleocapsid structure of paramyxoviruses can be found in [37,46,47].

### 3.2. Nucleocapsids of Segmented, -ssRNAv

The order *Bunyavirales* (comprising the families *Hantaviridae*, *Peribunyaviridae*, and *Arenaviridae*, among others) and the order *Articulavirales* (particularly the family Orthomyxoviridae) encompass most of the known segmented -ssRNAv. The genomes of these viruses are fragmented into several pieces and are, therefore, sometimes referred to as *Multinegavirales* [48]. Genome partitioning implies the presence of multiple RNPs, as many as the number of RNA segments, which can offer evolutionary advantages such as an extra degree of freedom for recombination, facilitating the mixing of genome segments to create hybrid offspring in case of infection by different strains (antigenic shift) [49]. However, this advantage has a trade-off, since the formation of infectious particles depends on greater regulation in order to ensure each progeny virion receives a copy of each genomic segment.

In segmented viruses, the largest gene always corresponds to RdRp and another of the segments encodes the N protein. Another common feature is that the 3’ and 5’ ends of the RNA segments are complementary, pairing to form a small double-stranded RNA fragment that serves as the binding site of RdRp, thus forming an intrinsic part of the RNP. In addition, these viruses lack the phosphoprotein that acts as a cofactor for transcription and replication in the *Mononegavirales* [50]. Finally, these RNPs are generally much more flexible in nature than those of the non-segmented -ssRNAv, a characteristic that is closely related to their function.

#### 3.2.1. Bunyavirales 

*Bunyavirales* constitute the largest order of segmented -ssRNA viruses. Their virion particles are pleomorphic, roughly spherical, enveloped, and contain a genome comprising three segments: L (large) encoding the RdRp; M (medium) encoding the glycoprotein precursor; and S (small) encoding the N protein [51]. In addition to these essential proteins, many viruses of the *Bunyavirales* encode non-structural proteins within the S and M segments that are often key virulence factors [52]. The terminal nucleotides of each genome segment are complementary, pairing to form non-covalent circular RNAs and, therefore, closed RNPs [26]. 

##### Hantaviridae: Hantaan Orthohantavirus (HTNV)

HTNV is the prototypic member of the *Hantaviridae* family, and it can cause hemorrhagic fever with renal syndrome in humans. Its genome is divided into three RNPs that encode four proteins [53]. The first structural studies on native HTNV virions were performed by cryoET, finding rod-shaped structures in the interior that were assigned to RNPs [54]. These RNPs are approximately 10 nm in diameter and arranged in parallel bundles, forming pairs, triplets, or even larger arrangements. More than three “rods” were detected within many of the virions analyzed, so the authors assumed that the RNPs had undergone 180º bends. Due to the limited resolution (7.5 nm), the authors could not determine the length of the RNPs present inside the virion, nor could they rule out that more than one copy of the RNPs could have been packaged into the virion [54].

Later, the full-length N protein was expressed in insect cells and found to spontaneously form a rigid helical structure with a diameter of 110 Å [55] (Figure 4A,B), which matches that observed by cryoET within virions [54]. Determination of the structure of these helices at 3.3 Å resolution revealed a unique structure in which there are only 3.6 N monomers per turn and the RNA binding groove is oriented towards the inside of the RNP, contrary to those previously described which were facing outward [55]. The RNA present in the structure belonged to the host cell, again indicating a non-specific sequence interaction which is usual for N proteins. Additionally, the first 70 amino acids of the protein were not visible in the final map, indicating that they are not ordered. The RNP structure is compatible with the previously proposed model that suggested the N-proteins first trimerize around the viral RNA [56], and then gradually assemble into longer multimers. The combination of this structural data with the extensive previous data allowed the authors to hypothesize a mechanism for transcription and replication [55].

##### Peribunyaviridae: Bunyamwera Orthobunyavirus (BUNV)

BUNV is the prototypic virus of the *Peribunyaviridae* family, and it possesses three genomic RNA segments (961, 4458, and 6875 bases) distributed in three RNPs. 

Characterization by electron microscopy of isolated BUNV RNPs differentiated the three RNPs by size, with closed helical segments of about 150, 650, and 1000 nm in the perimeter [57]. CryoEM studies using SPA and cryoET determined that the RNPs are extremely flexible helical structures with approximately 4 NPs per turn. The obtained reconstruction at a resolution of about 18 Å (Figure 4C) was complemented with atomic force microscopy analysis to determine the absolute hand of the structure. Docking of the NP monomer atomic structure was used to generate a model of the RNP (Figure 4D), which placed the genomic RNA on the inner face of the helix, where it is almost completely covered [57] (Hopkins mBio 2022). This is consistent with the model previously proposed for *Bunyavirales* segment packaging [58].

##### Nairoviridae: Crimean-Congo Hemorrhagic Fever Orthonairovirus (CCHFV)

CCHFV is the most relevant member of the *Nairoviridae* family from a human health perspective. *Nairoviruses*, and specifically CCHFV, possess the longest genomes of the *Bunyavirales*, with lengths of 12160, 5366, and 1672 nt in the L, M, and S segments, respectively, which confers the longest RNPs, and also the largest RdRp and NPs [26,59]. On the other hand, it has been demonstrated that CCHFV NP has endonuclease activity when found as a monomer, being able to degrade double- and single-stranded DNA, but not RNA [59].

Overexpression of CCHFV NP in *E. coli* yielded ring-shaped oligomers with diameters ranging from 120 to 220 Å, containing from four to seven NP molecules and bound to host cell RNA fragments [60]. The dominant arrangement, the pentameric ring, was selected to perform a cryoEM reconstruction that reached a final resolution of about 10 Å, and docking of the available NP atomic structure was performed (Figure 4F). In the obtained model, the RNA binding groove overlaps with the active site of the NP endonuclease, supporting the hypothesis of inhibited NP enzymatic activity only in the oligomeric state of the RNP. In this work, native RNPs isolated from virions were also imaged, in which most had a flexible helical architecture with a diameter of approximately 50 Å (Figure 4E). However, in some regions these RNPs showed a diameter of about 120 A, compatible with an arrangement very similar to that found in the rings obtained by NP overexpression [60].

**Figure 4 viruses-15-00653-f004:**
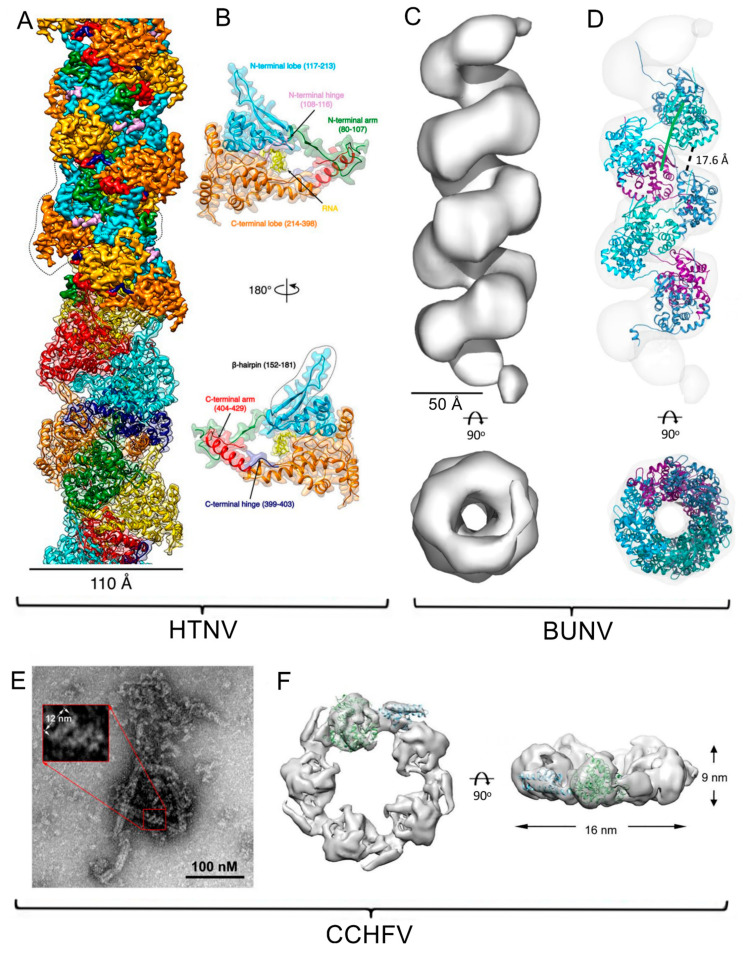
CryoEM structure of different RNPs of viruses of the order *Bunyavirales*. (**A**) Structure of an RNP-like particle obtained from overexpression of Hantaan orthohantavirus (HTNV) nucleoprotein (NP) interacting with host cell RNA. The upper part of the cryoEM map is colored by NP domain; cyan represents the N-terminal lobe, pink the N-terminal hinge, green the N-terminal arm, orange the C-terminal lobe, red the C-terminal arm, navy blue the C-terminal hinge and yellow the RNA, while the lower part is colored by NP protomer. The NP surrounded by dotted lines corresponds to Figure 4B orientation. (**B**) Two views of the NP domains of a monomer extracted from the helical assembly (color code as in A) [55]. (**C**) CryoEM structure of the Bunyamwera orthobunyavirus (BUNV) RNP at 13 Å resolution. (**D**) Pseudoatomic model of the BUNV RNP obtained after flexible fitting of an NP model; distance between rungs of the helix is indicated [57]. (**E**) Negative staining EM image of native Crimean-Congo hemorrhagic fever orthonairovirus (CCHFV) RNPs extracted from virions, showing the high flexibility of these complexes. (**F**) SPA cryoEM 3D reconstruction of a CCHFV NP oligomer obtained from overexpression, with the crystal structure (PDB 3U3I) docked in the map [60].

NP-RNA complex structures have been solved for other viruses within the order of *Bunyavirales* (e.g., La Crosse Orthobunyavirus) [61] and reviews specific to their nucleocapsid and RNP structures are available [62].

#### 3.2.2. Articulavirales

The order *Articulavirales* contains the family *Orthomyxoviridae*, which comprises influenza A, B, C, and D viruses as well as related viruses such as infectious salmon anemia virus and thogotovirus. Virions are pleomorphic although essentially spherical (80 to 120 nm in diameter), and also filamentous in some cases. The genomes of this order are divided into 8 (for influenza A, B, and salmon anemia virus), 7 (for influenza C and D), or six (for thogotovirus) segments, with a corresponding number of RNPs. The conserved ends of the genomic RNA segments are complementary, pairing to form a small fragment of double-stranded RNA that serves as the binding site of RdRp, which also has promoter activity.

##### Orthomyxoviridae: Influenza A Virus (IAV)

IAV is undoubtedly the most important orthomyxovirus from a human health perspective. Its nucleocapsid is composed of eight RNPs with segments ranging in length from 890 to 2340 bases, for a total size of about 13.5 kb that encodes 12 to 18 proteins, depending on the strain. The RNPs of IAV have been extensively studied since their discovery and isolation [63], attempting to determine their function, structure, and how they are packaged within the virion. The general structure described in early microscopy works showed that IAV RNPs are rod-like complexes, consisting of a double helix with a closing loop at one end and the viral polymerase at the other. Their diameter is approximately 14 to 18 nm with a length that varies based on the size of the RNA segment it contains (from 40 to 150 nm) [63]. 

Due to the great flexibility of native IAV RNPs, the earliest structural studies using image processing were performed on recombinant mini replicons that contained the polymerase, some copies of the NP, and a genomic RNA with conserved 3′ and 5′ ends but whose length was only about 240 to 350 nt [64,65]. CryoEM was used to show that the mini replicons were rings containing nine NP monomers (12 Å resolution), with the polymerase (18 Å resolution) attached via interaction with the paired ends of the genomic RNA [66]. The structure of full-sized IAV RNPs was later determined, either from native RNPs isolated from virions [67] or RNPs obtained by overexpression in mammalian cells [68]. These data confirmed that RNPs are organized in an antiparallel double helix, with a major and a minor groove resulting from the folding of the RNA/NP chain on itself due to the interaction of the 3’ end with the 5’ end (Figure 5A,B). The polymerase is located at one end of the double helix, interacting with the small double-stranded segment formed by base-pairing between approximately 12 terminal nucleotides at each end of the RNA segment. The other end is a closing loop formed by the region where the RNA-NP strand turns on itself to form the double helix. Arranz et al. also performed cryoET of IAV virions to show the nucleocapsid containing the packed RNPs. Tomograms exhibited RNPs in the virion with the same double helix structure that had been determined in isolation and also showed that they were arranged approximately in parallel to each other, forming bundles (Figure 5C) [68].

A number of studies were also carried out using electron microscopy and conventional tomography, looking at how the eight RNPs are packaged to form the complete nucleocapsid. In these works, the predominant packaging arrangement was a “7 + 1” type configuration, in which one RNP is positioned as the central axis of the nucleocapsid and is surrounded by the seven other RNPs to form a bundle of parallel molecules [69,70,71].

**Figure 5 viruses-15-00653-f005:**
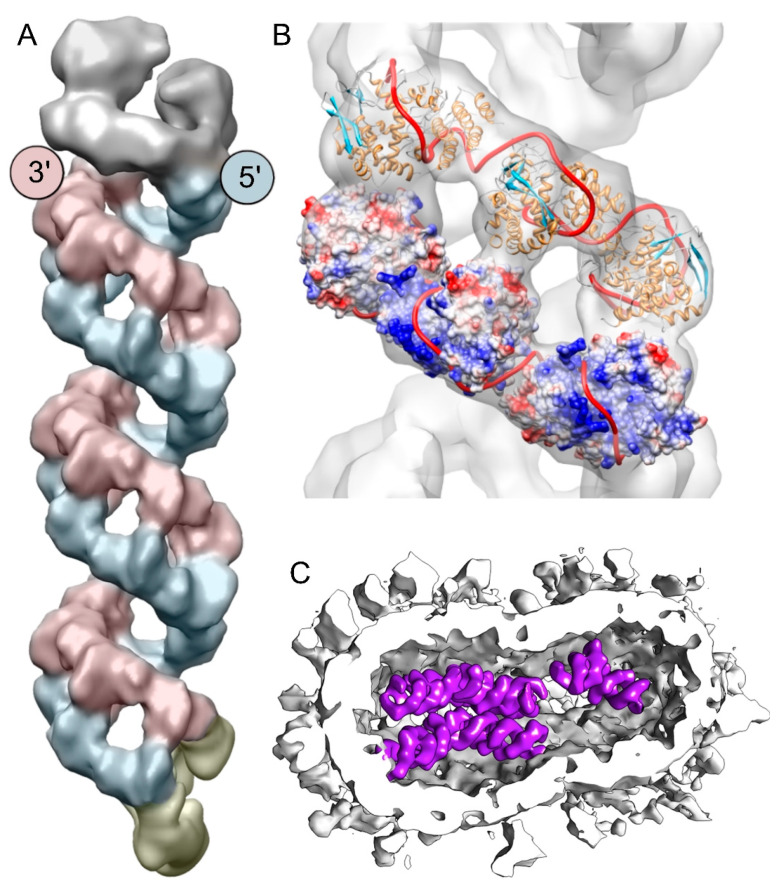
3D structure of the native RNP of influenza A virus (IAV) and its arrangement in the virion. (**A**) Model of the segment 8 RNP [67]. The RdRp is shown in gray, the antiparallel nucleoprotein (NP) strands are shown in red and blue, and the closing loop is in yellow. The positions of the 3′ and 5′ ends are defined as described in [72]. (**B**) Docking of the NP monomer (PDB:2IQH) in the helical region of the RNP. In the lower strand NP, monomers are represented as electrostatic potential surfaces, and in the upper strand as ribbons; the modeled RNA template is depicted as a red thread. (**C**) Segmented tomogram of an IAV virion in which subtomogram-averaged RNP segments (in purple) are placed back on the position, with orientation computed in the alignment [67].

The best resolution achieved by cryoEM SPA in the early studies was around 18 Å [67], and this low/limited value was due to the high flexibility of IAV RNPs. To study the source of such flexibility, an extensive 3D classification of cryoEM images was performed. It was seen that the relative position of the NPs on one strand of the double helix was not fixed with respect to the NPs on the opposite strand, but rather could vary widely, producing both displacements and twists of the NP monomers (Figure 6A,B) [10]. Detailed analysis of this variability led to the discovery of the action mechanism by which RNPs exert their transcription and replication functions. To determine the RNP dynamics during RNA synthesis, electron microscopy images were taken during the transcription process, revealing that the RNP undergoes a sliding movement of one strand over the other. This movement allows the polymerase to move through the RNA template present in the RNP without disrupting the double helix structure (Figure 6C), giving the transcription mechanism the processivity - i.e., the ability to be made repetitively - that had been described biochemically [10]. This study also determined by cryoEM the structure resulting from treating RNPs with the drug nucleozin, an inhibitor of transcription and replication of IAV, which alters the organization of the double helix to prevent displacement between the two strands, therefore inhibiting the process of RNA synthesis.

## 4. Structure of the Nucleocapsid of +ssRNAv

Viruses whose genetic material is +ssRNA have the advantage that their genomes can behave directly as functional mRNA, being in many cases capped and polyadenylated in a manner similar to cellular mRNAs [73]. Although every known +ssRNAv carries a gene for an RdRp that is used in genome replication and transcription, most do not encapsidate this protein into the virion. Viral RNA replication cannot begin after infection until the genomic RNA is translated to produce polymerase and additional replication factors, making protein synthesis the first synthetic event in the life cycle. This requirement for direct translation of the viral genome as the first event after infection is one of the major differences between positive- and negative-polarity ssRNAv. Consequently, the original infecting genome has two functions: it serves as mRNA for the synthesis of new proteins, and it is also the template for progeny viral RNAs [74]. Thus, the genomic RNA that is condensed within an RNP must first be disassembled to expose the RNA so that it can be translated by the ribosome, contrary to what we have described for -ssRNAv, in which the RNP retains its structure at all times. However, newly synthesized RNPs appear in the cell at the end of the infection to be packaged into the virion.

Families of +ssRNAv include the *Picornaviridae*, *Flaviviridae*, *Togaviridae*, *Hepeviridae*, *Coronaviridae*, *Arteriviridae*, and *Toroviridae*, among many others. Most families present a non-segmented genome, but some families exhibit segmentation, such as viruses from the family *Nodaviridae* (bisegmented).

The +ssRNAv show greater variation in morphology and structural features between them compared with the -ssRNAv (e.g., the presence or absence of an envelope). The most common nucleocapsid morphology for +ssRNAv is icosahedral, in contrast with the characteristic helical conformation present in most of -ssRNAv. However, viruses within the *Coronaviridae* family are distinctive due to their helical nucleocapsids and it is on these that we will focus on in this review.

Coronaviruses (CoVs) (family: *Coronaviridae*; order: *Nidovirales*) are enveloped +ssRNAv that cause mostly respiratory and gastrointestinal diseases in avian and mammalian species [75]. The non-segmented CoV genome (26-32 kb) is the largest of all RNA viruses, and associates tightly with the N protein to form RNPs with a helical structure [76]. Members of the *Betacoronavirus* genus, including severe acute respiratory syndrome coronavirus I and II (SARS-CoV-1 and -2) and Middle East respiratory syndrome-coronavirus (MERS-CoV), have been studied extensively because of their important implications for human health.

The first studies of negatively stained RNP complexes extracted from disrupted CoV virions revealed thread-like coil structures with diameters of 9 to 16 nm and a hollow interior of approximately 3 to 4 nm [77]. Neumann et al. performed one of the first cryoEM studies to examine the ultrastructure of SARS-CoV-1, feline coronavirus (FCoV), and murine hepatitis virus (MHV), including their nucleocapsids. The RNPs of these viruses were shown as an array of 5 by 6 nm oval densities arranged in an approximately paracrystalline order [78]. Since then, several cryoEM 3D reconstructions of the CoV nucleocapsid have been achieved at different resolutions. A cryoET study of MHV by Bárcena et al. showed the RNP densely packed under the envelope but, despite the overall poor organization of the nucleocapsid, there were small structures (~11 nm diameter) with quasi-circular density profiles enclosing an empty space of approximately 4 nm in diameter. All these data points to the fact that the CoV RNP appears to be folded extensively in on itself, assuming a compact structure that tends to closely follow the membrane in the vicinity of the envelope [79]. Later, MHV RNPs were also studied in isolation after detergent treatment of virions, revealing two arrangements: a loose filament structure; or a compact flower-like assembly, also pointing to a possible helical arrangement of the N proteins [80].

More recently, the SARS-CoV-2 pandemic has prompted a large number of structural studies by cryoEM both in SPA and cryoET. However, most of them have been devoted to the investigation of the envelope spike (S) protein, and very few to the RNP structure. One reason for the relative lack of attention is the complexity of the nucleocapsid organization and the enormous flexibility of the CoV RNP, which make the studies extremely challenging. Some of these works have been carried out using the techniques of correlative cryomicroscopy and lamella production for cryoET, which make it possible to analyze the structure of newly synthesized RNPs within the host cell before they are packaged into the virus. Thus, Klein et al. characterized the viral replication compartment and the early budding process (which contains progeny RNPs) using in situ cryoET on cryo-focused, ion beam-milled lamellae or whole-cell cryoET of various SARS-CoV-2-infected cell lines, fixed previously with a mixture of paraformaldehyde and glutaraldehyde [81]. In this work, the authors also studied extracellular virions using subtomogram averaging, showing that RNPs are distinct cylindrical assemblies. They proposed that the genome is packaged around multiple RNP complexes, similar to a beaded necklace, which allows the incorporation of the large genome into the virion while still maintaining high steric flexibility between the RNPs [81].

Yao et al., also using cryoET, performed an analysis covering a large dataset of SARS-CoV-2 virions [82]. Virions were obtained from an early virus strain, propagated in Vero cells, and fixed with paraformaldehyde prior to their visualization, which revealed enveloped particles with two characteristic shapes, ellipsoidal and spherical (Figure 7A,B). Tomograms showed the RNP organized in the nucleocapsid in approximately spherical blobs with a diameter of about 16 nm, and a significant number of these blobs were located close to the membrane. An in-depth study of the arrangement of the blobs showed two main conformations: one close to the membrane, in the form of ‘‘eggs in a nest’’ (or hexons); and another in the form of a “pyramid” (or tetrahedron) for regions away from the membrane (Figure 7A). Moreover, the majority of the hexons came from spherical virions, whereas more tetrahedrons came from ellipsoidal virions (Figure 7B). Subtomogram averaging of the blobs revealed a more detailed structure with a resolution of 13 Å, showing a G-shaped architecture 15 nm in diameter and 16 nm in height that allowed the authors to propose a pseudo atomic model of the RNP basic unit (Figure 7C). This structure is quite similar to that described by Klein et al. [81], and also to the conformation shown for the Chikungunya virus RNP, a +ssRNAv of the *Togaviridae* family, also studied by cryoET [83].

Finally, Calder et al. performed another cryoET study on different SARS-CoV-2 variants (Wuhan, Alpha, Beta, and Delta) in which most of the viral particles had an oblate ellipsoid shape with a smaller radius of about 40 nm and a larger radius of about 100 nm, which matched with the nucleocapsid shape. The RNPs appeared to be arranged in a two-layer cylindrical assembly with an average height of 22 nm, and the blobs composing the RNP had a very similar structure to those described previously [81,82], with a diameter of around 15 nm [84].

## 5. Concluding Remarks

RNPs/nucleocapsids are probably the most complex structures present in viruses, and they play a crucial role in the different stages of their life cycle. Processes such as genome encapsidation, transcription, replication, and packaging of genetic material to form progeny virions in the infected cell, all fundamental to viral proliferation, are entirely governed by RNPs. The performance of these key functions brings to the forefront the interest in the study of the RNP’s structure-function relationship as one of the most important lines in the fight against these pathogens.

X-ray crystallography has traditionally been the mainstay of structural determination, as it continues to serve for atomic-resolution structural determination of many of the proteins that make up viral RNPs/nucleocapsids. However, since most RNPs are highly flexible complexes, a characteristic that is inherent to their function, the applicability of X-ray methodology is limited. CryoEM, and its subsequent development in what has been called the revolution of the resolution, has moved to become the key piece for the study of these viral complexes since it avoids the step of obtaining crystals. On the other hand, cryoEM is perfectly suited to the study of dynamic processes carried out by RNPs, due to the technique’s ability to isolate reaction intermediates that would not be observable otherwise. Furthermore, cryoET has brought an enormous leap in the determination of the ultrastructure of many viruses, especially those with lipid envelopes. Moreover, the new techniques of correlative cryomicroscopy and direct tomography on infected cells are opening a new path for the study of the events that occur in the cell over the entire infection process. In conclusion, the new combination of structure determination techniques made possible by cryoEM is set to revolutionize structural virology in the near future.

## Figures and Tables

**Figure 6 viruses-15-00653-f006:**
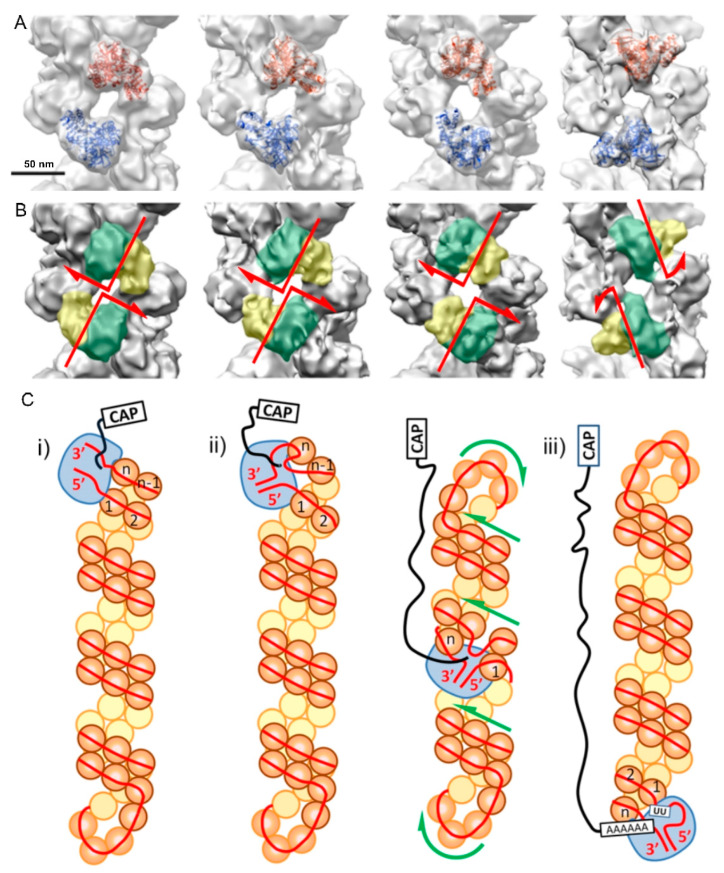
3D reconstructions of different conformations of the helical part of Influenza A virus (IAV) RNPs and the transcription process mechanism. (**A**) The docking of the nucleoprotein atomic structure (PDB:2IQH) is shown on the opposite strands in red and blue. Scale bar, 50 Å. (**B**) The nucleoprotein (NP) head and body domains are outlined in yellow and green, respectively, showing the variation of the relative position of two NP monomers from the opposite strands. Red arrows indicate the direction of displacement. (**C**) Scheme of the RNP during the different transcription steps: (i) initiation; (ii) elongation; (iii) polyadenylation. Green arrows indicate the relative movement between the strands [10].

**Figure 7 viruses-15-00653-f007:**
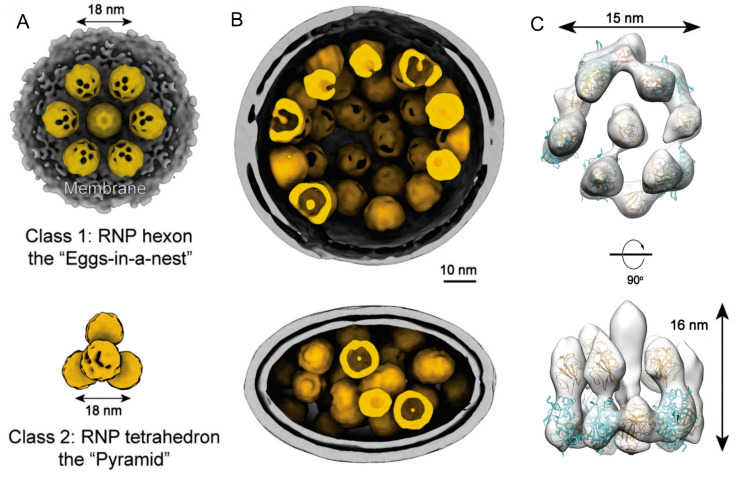
The native assembly and tentative model of SARS-CoV-2 RNPs [82]. (**A**) Ultrastructure of the RNP hexon and tetrahedron assemblies. At the top, seven RNPs are packed against the viral envelope (gray), forming an “eggs-in-a-nest” hexagonal assembly. At the bottom, four RNPs are packed as a membrane-free tetrahedral assembly. (**B**) Representative projection of RNP hexons assembling into a spherical virus (top) and tetrahedrons into an ellipsoidal virus (bottom). (**C**) 13.1 Å-resolution RNP showing a reverse G-shaped architecture, measuring 15 nm in diameter and 16 nm in height. N-terminal (6WKP) and C-terminal (6WJI) domains of N are fitted into the map in each head-to-tail reverse L-shaped density.

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
