# Peer review of "CryoEM of Viral Ribonucleoproteins and Nucleocapsids of Single-Stranded RNA Viruses"

_viruses, 2023, doi:10.3390/v15030653_

Round 1

Reviewer 1 Report

CryoEM of viral ribonucleoproteins and nucleocapsids

 This is a very nice review about the nucleocapsids of negative stand RNA viruses, for the viruses with a long RNA genome and for the segmented viruses. I have a few comments for the figures but I have a problem with 4: Structure of the nucleocapsid of the +ssRNAv. This group of viruses don’t have “nucleocapsids” or “ribonucleoprotein, RNP”. These viruses have RNA and it must have a kind “structure” but very different from the –ssRNA viruses and for that, I would take this part of this review away. If the authors want a short review on the RNA of coronaviruses, they really try not using the words “nucleocapsid”, “RNP” etc. As is written now, the authors talk about; “eggs”, “blobs” but not about a nucleocapsid! I looked at some of the papers on the structure of coronaviruses, and each paper see different forms. Like many colleagues, we have to find a structure of the RNA after opening the particles and we have found this like found in the other papers on the ref list but we have found never a “nucleocapsid”. In a way, it is sad finishing a very nice review about the structure of the RNA of coronaviruses.

 Figure 1; The helix shown in A is different of the helix in D, they don’t have a same protomer per turn of the helix? The same for E and H. Please mention the number of protomer per turn.

Fgure 4; Show the size of the monomer in B, must looks bigger that the monomers in A. E is really too small, I saw almost nothing! The ring in F left has not the same size as in right.

Figure 7; I think that you should take this figure and the text out.

Author Response

We are very grateful for the reviewer's comments which help us to improve the quality of the manuscript.

Comments on Figure 1. The number of subunits per turn has been added in all cases.

Comments on Figure 4. The figure has been corrected following the reviewer advices.

With regard to the comment concerning the +ssRNAv section and figure 7, we do not understand what the reviewer is referring to specifically. We agree that the organization of the nucleocapsid of these viruses is not as ordered as that of the -ssRNAv. However, we included this section in the review because in the scientific literature that we have handled, and which is reflected in the references, it always appears that the genome of the +ssRNAv is organized in RNPs. For example, in the first full description of SAR-CoV-2 virus (Yao et al., 2020, Cell), the term RNP is included almost a hundred times throughout the article, including the section describing the organization of the genome, entitled "Architecture and Assembly of RNPs in Intact Virions". The term RNP and nucleocapsid is also used, even for the packaging of the viral genome within the cell prior to budding, in the paper by Klein et al. (Klein et al., 2020, Nat. Commun). Other articles include the term RNP in the title, as in Clader et al. who title it "Electron cryotomography of SARS-CoV-2 virions reveals cylinder-shaped particles with a double layer RNP assembly" (Calder et al., 2022, Commun. Biol.). The term RNP and nucleocapsid is also used for the other coronaviruses mentioned in the review, such as Murine Hepatitis Virus (Bárcena et al., 2009, PNAS), SARS-CoV-1 and feline coronavirus (Neuman et al. 2006, J.Virol.).

For all these reasons we respectfully think that the terms RNP and nucleocapsid can be used for +ssRNAv as well and, due to the great interest that these viruses have nowadays for society, we believe that the section should be kept in the text.

Reviewer 2 Report

Modrego et al. submit to Viruses a review entitled "CryoEM of viral ribonucleoproteins and nucleocapsids".

The review covers recent works on the structural biology of viral ribonucleoproteins and nucleocapsids and most particularly with the use of the new CryoEM (SPA and CryoET) methods. The work is very useful, well written and interesting. The figures are rich and contains a lot of information, however we notice that some information are lacking in the legend as described in the comments below.

Comments

The review deals exclusively with the single strand viruses (ss viruses), numerous cryoEM studies covering lentiviruses or Hepatitis B viruses are not included while the terms of capsids, nucleocapsids and ribonucleoproteins are currently used with these latter viruses that are not ss viruses. Perhaps the authors should precise in the title that the review deals exclusively with ribonucleoproteins and nucleocapsids of ss viruses.

The paragraph 2. "The single-stranded RNA viruses and riboncleoproteins" is quite poor in references

Ref 6 no editor of the book ?

l108 non-segemented

Figure 2 A the RNA(red) is not mentioned in the legend and the observable yellow color is not mentioned

Figure 3 C displays two different views of the Nucleocapsid like particle but it is not explained what are these two views, why they are different

Figure 4A in the legend that describes the NP domains, C-terminal -arm of the nucleoprotein protomer is not mentioned  protomer (red)

Fig 4E is never mentioned in the text

Legend of figure 5 B we think that the NP in the lower strand are repsented with their electrostatic potential surface but the term electrostatic is not present

Figure 7 C the structures that are superimposed on the cryoEM envelope are domains of the N protein that are not commented, and nod described

Author Response

We appreciate the reviewer's constructive comments.

Certainly, the reviewer's assessment is correct and the article is focused on ssRNAv, therefore we have modified the title following his advice to "CryoEM of viral ribonucleoproteins and nucleocapsids of single-stranded RNA viruses".

As suggested by the reviewer we have also expanded the number of references in paragraph 2 and it now contains references 5 to 10.

As requested by the reviewer, the editor (Robert Lamb) has been added to reference 6 (now in the new version of the manuscript position number 11).

In line 108 (now 109) the typographical error has been corrected.

All the corrections suggested by the reviewer for the figures had been made. The reference to figure 4E has been include in the line 341 of the text.

Reviewer 3 Report

Authors provide a comprehensive review of studies focusing on the cryoEM structures of viral components with a primary focus on single stranded viruses and involved ribonucleoproteins. Given that single stranded RNA viruses have been implicated to contribute towards diversity and on the other hand pose a major threat to human health, detailed understanding of the mechanisms involved will open further doors for developing novel approaches to tackle the posed challenges. The review article begins by describing what single stranded genomes are and how genome polarity affects the viral replication cycle. Authors provide a comprehensive overview of the studies investigating single stranded RNA viruses focusing on ribonucleoprotein structures, polarity, structure of nucleocapsid. Authors then individually discuss different groups of viruses providing the overview on their individual pathological and then structural context.

Overall, the article is well structured and well written and ready for publication except for following minor point.

1.     The resolution for figure 2 and figure 3 should be improved.

Author Response

We very much appreciate the reviewer's positive comments. We have corrected the resolution of figures 2 and 3 as far as possible.